# Bifunctional Nitrogen and Fluorine Co-Doped Carbon Dots for Selective Detection of Copper and Sulfide Ions in Real Water Samples

**DOI:** 10.3390/molecules27165149

**Published:** 2022-08-12

**Authors:** Yutian Zeng, Zhibin Xu, Jiaqing Guo, Xiantong Yu, Pengfei Zhao, Jun Song, Junle Qu, Yu Chen, Hao Li

**Affiliations:** Center for Biomedical Optics and Photonics (CBOP) & College of Physics and Optoelectronic Engineering, Key Laboratory of Optoelectronics Devices and Systems of Ministry of Education/Guangdong Province, Shenzhen University, Shenzhen 518060, China

**Keywords:** carbon dots, fluorescent probe, copper detection, sulfide detection

## Abstract

Copper ions (Cu^2+^) and sulfur ions (S^2−^) are important elements widely used in industry. However, these ions have the risk of polluting the water environment. Therefore, rapid and quantitative detection methods for Cu^2+^ and S^2−^ are urgently required. Using 2,4-difluorobenzoic acid and L-lysine as precursors, nitrogen and fluorine co-doped dots (N, F-CDs) were synthesized in this study via a hydrothermal method. The aqueous N, F-CDs showed excellent stability, exhibited satisfactory selectivity and excellent anti-interference ability for Cu^2+^ detection. The N, F-CDs, based on the redox reactions for selective and quantitative detection of Cu^2+^, showed a wide linear range (0–200 μM) with a detection limit (215 nM). By forming the N, F-CDs@Cu^2+^ sensing platform and based on the high affinity of S^2−^ to Cu^2+^, the N, F-CDs@Cu^2+^ can specifically detect S^2−^ over a linear range of 0–200 μM with a detection limit of 347 nM. In addition, these fluorescent probes achieved good results when used for Cu^2+^ and S^2−^ detection in environmental water samples, implying the good potential for applications.

## 1. Introduction

With industrial development, environmental pollution caused by industrial byproducts has become a major hazard to human health, owing to the enrichment of toxic substances in the human body. Copper is an essential element in industrial production and widely used in electrical, manufacturing, metallurgical, and petrochemical industries [1,2]. Copper ion (Cu^2+^) is also an essential trace element in human beings, animals, and plants, playing a crucial role in bone formation, cellular respiration, and connective tissue development [3,4,5]. However, many diseases are related to excessive amounts of Cu^2+^, such as gastrointestinal disturbances, liver and kidney damage, Parkinson’s disease, and Alzheimer’s disease [6,7,8,9]. Cu^2+^ ions are listed as pollutants, and their concentration in drinking water should not exceed 20 μM, according to the United States Environmental Protection Agency [10]. Although sulfide is an important chemical used in catalysis, batteries, dyeing, and agriculture, its ions (S^2−^) are highly toxic to living organisms [11,12,13]. Furthermore, the hydrolysis product of S^2−^ ions (H_2_S) is considered to be involved in many diseases, such as diabetes [14], Down’s syndrome [15], and liver cirrhosis [16].

Considering the wide application and toxicity of Cu^2+^ and S^2−^ to humans, the quantitative detection of these ions in the environment requires urgent solutions. Current studies have applied techniques, such as ion chromatography [17] and spectrophotometric titration [18], for Cu^2+^ detection; capillary electrophoresis [19,20,21,22] and chemodosimeter are used [23] for S^2−^ detection. However, these methods require multi-step preprocessing, and they have a high cost and low selectivity. To overcome these shortcomings, fluorescent methods for detecting Cu^2+^ and S^2−^ ions, including quantum dots [24], metal nanoclusters [25], and organic-molecule fluorescent probes [26], have received increasing attention. However, these luminescent methods utilize organic chromophores that are difficult to prepare and cannot achieve multiple-ion detection, limiting their practical applications. Therefore, easily synthesized probes, with specificity and accuracy for Cu^2+^ and S^2−^ detection, are required.

In recent years, carbon dots (CDs) have attracted significant interest due to their superior properties, such as excellent hydrophilicity, remarkable photostability, high sensitivity, and tunable selectivity. With the continuous optimization, CDs have been extensively applied to various fields, such as detection [27], biosensing [28], batteries [29], catalysis [30] and biomedicine [31]. Although CDs offer many advantages and are increasingly used in ion detection, including Cu^2+^ and S^2−^ ions, only a few studies have reports on bifunctional CDs that can detect Cu^2+^ and S^2−^ ions quantitatively and selectively.

In this study, nitrogen and fluorine co-doped dots (N, F-CDs) were synthesized via a hydrothermal method. The specificity of the N, F-CDs for Cu^2+^ detection was demonstrated by reacting with a variety of ions. Furthermore, these probes can apply to S^2−^ detection by forming the N, F-CDs @Cu^2+^ sensing platform. In addition, the sensing mechanisms of detecting Cu^2+^ and S^2−^ were explored through a series of experiments. Therefore, a bifunctional fluorescence sensor was constructed and provided good results when applied to quantitatively detect Cu^2+^ and S^2−^ in real water samples.

## 2. Results and Discussion

### 2.1. Characterization of N, F-CDs

The morphology and size dispersion of the synthesized N, F-CDs were characterized by TEM image (Figure 1a). The shape of the N, F-CDs were mostly spherical with a uniform dispersion. Figure 1b shows the particle size distribution of the N, F-CDs obtained, resulting from 100 particles, which ranged from 1.25 to 6 nm, with an average size of 3.32 nm. The incorporation of different elements and the functional groups on the surface of N, F-CDs were confirmed by the FT-IR spectrum. The broad absorption peak at 3328 cm^−1^ is assigned to C–H/O–H stretching, while the peaks at 1632, 1381, and 1141 cm ^−1^ are ascribed to the C=C, C–N, and C–F stretching, respectively. All the chemical bonds mentioned above indicate the successful doping of different elements into the N, F-CDs.

To confirm the presence of functional groups and analyze the composition of the N, F-CDs, XPS profiles were obtained. The full XPS profiles of the N, F-CDs (Figure 2a) four predominant peaks at 285, 401, 531, and 687 eV, corresponding to C1s, N1s, O1s, and F1s, respectively. The N, F-CDs contained 66.3% of carbon, 11.2% of nitrogen, 15.5% of oxygen, and 7% of fluorine. The high-resolution spectrum of C1s (Figure 2b) exhibited four peaks at 284.97, 286.0, 287.2, and 288.7 eV, which are assigned to C–C/C=C, C–N/C–O, C=O, and C–F bonds, respectively. The high-resolution N1s spectrum (Figure 2c) showed three peaks at 399.1, 399.78, and 401.3 eV, which are attributed to the pyridinic, pyrrole, and graphitic N atoms, respectively. The high-resolution O1s spectrum (Figure 2d) demonstrated two peaks at 531.18 and 532.88 eV, which are attributed to C=O and O=C–O bonds, respectively. The high-resolution F1s spectrum (Figure 2e) showed two peaks at 685.2 and 687.2 eV, which are assigned to C–F and covalent C–F groups, respectively. The FT-IR and XPS results confirmed the incorporation of nitrogen and fluorine into the N, F-CD structure. In addition, the N, F-CDs contain various functional groups that are available for reaction with ions and improve hydrophilicity.

UV-vis and fluorescence spectra were analyzed to determine the optical properties of N, F-CDs (Figure 3a). The UV-vis absorption of N, F-CDs (blue line in Figure 3a) exhibited an absorption peak at 230 nm, corresponding to the n–π* transition of the C=C bond. The maximum emission wavelength (red curve in Figure 3a) was located at 505 nm under a maximum excitation wavelength of 468 nm (black curve in Figure 3a). The excitation-emission map of the N, F-CDs aqueous solutions (Figure 3b), in the excitation range of 350 to 500 nm, indicated the maximum emission wavelength did not shift with changes in excitation wavelength, which is a typical excitation-independent fluorescence phenomenon. The quantum yield of the N, F-CDs at room temperature (25 °C) was 38.65%.

### 2.2. Fluorescence Stability of N, F-CDs

The chemical and optical stabilities of the N, F-CDs were investigated in terms of ionic strength, pH, temperature, and irradiation time [32,33,34]. Figure 4a illustrates the effects of different NaCl concentrations on the fluorescence intensities of N, F-CDs, exhibiting the good stability of the N, F-CDs in strong ionic strength. Figure 4b shows a comparison of the N, F-CD fluorescence intensities in increasing pH levels from 1 to 14. The N, F-CDs exhibited high optical stability under weakly acidic to weakly alkaline conditions (pH = 3.0–11.0). Strong acidic conditions (pH = 1.0–3.0) cause fluorescence quenching, whereas strong alkaline conditions (pH = 14.0) gradually weaken the fluorescence intensities until almost no fluorescence is observed. Thus, the N, F-CDs were remarkably stable over a wide range of pH values, making them potentially suitable for applications. Figure 4c illustrates the effect of temperatures on the fluorescence intensities, exhibiting a slight intensity decrease within a wide temperature range (20–90 °C). Figure 4d shows a comparison of N, F-CD fluorescence intensities when exposed to UV light for 1.5 h. A slight variation was observed throughout the exposure time, indicating the good photostability of the N, F-CDs. These results imply that the N, F-CDs are photostable and have good potential for ion detection in biological systems and bioimaging.

### 2.3. Detection of Cu^2+^ by N, F-CDs

Figure 5a shows the effect of different ions on fluorescence intensity. Only the addition of Cu^2+^ reduced the fluorescence intensity, while others have almost no or few effects on the fluorescence. Furthermore, an anti-interference experiment of metal ions on was performed to detect of Cu^2+^. Figure 5b illustrates that the addition of the metal ions to the Cu^2+^-quenched N, F-CDs did not affect the quenched fluorescence intensity, indicating the excellent anti-interference ability of the N, F-CDs for Cu^2+^ detection. Figure 4c shows the time evolution by Cu^2+^ ions toward the quenching of N, F-CD fluorescence intensity. The fluorescence intensity rapidly decreased after adding Cu^2+^ to the N, F-CDs at 2 min and remained stable for 10 min. Figure 5d illustrates the effect of Cu^2+^ concentration on the fluorescence intensity. The fluorescence intensity decreased 40% compared to the original N, F-CD fluorescence intensity when the concentration of Cu^2+^ was 500 μM. It should be noted that the maximum emission peak of the N, F-CDs did not shift throughout the fluorescence quenching, which suggests that Cu^2+^ reacts with the functional groups on the N, F-CD surface rather than occurs in the band structure. Figure 5e indicates the linear relationship between the ratio of the fluorescence intensity change (F_0_ − F)/F_0_ and Cu^2+^ concentrations from 0 to 40 μM and 40 to 200 μM; F and F_0_ are the emission intensities of the N, F-CDs in the presence and absence of Cu^2+^, respectively. The R_1_^2^ and R^2^ coefficient of 0.99531 and 0.99347, from 0 to 40 μM and from 40 to 200 μM, respectively, were estimated by linear fitting. The limit of detection (LOD) was determined to be 215 nM, which was calculated using the equation 3σ/k (σ represents the standard deviation of 3 blank measurements, and k is the slope of the linear calibration plot). Table 1 illustrates a comparison of the present study with previously reported systems. The results show that the N, F-CDs system has a wider linear range and a lower detection limit. Thus, the N, F-CDs can be applied to Cu^2+^ detection with high sensitivity.

### 2.4. Detection of S^2−^ by N, F-CDs@Cu^2+^

The detection properties of the N, F-CDs@Cu^2+^ sensor were verified through further experiments with anions. Figure 6a shows the fluorescence quenching properties of N, F-CDs@Cu^2+^ after adding individual anions. The fluorescence intensity was further quenched upon the addition of S^2−^. However, other anions showed no significant effect on the quenched fluorescence, indicating the selectivity of the N, F-CD@Cu^2+^ sensing platform. Figure 6b depicts the time evolution of the quenching of N, F-CDs@Cu^2+^ fluorescence intensity by S^2−^ ions. After adding S^2−^ for 30 s, the fluorescence intensity sharply decreased and remained stable for 10 min. Figure 6c illustrates the effect of S^2−^ concentration on the fluorescence intensity. The fluorescence was almost completely quenched to the original N, F-CD fluorescence intensity when the concentration of S^2−^ was increased to 500 μM. Figure 6d shows the linear relationship between the ratio of the fluorescence intensity change (F_0_ − F)/F_0_ and S^2−^ concentrations from 0 to 200 μM, where F_0_ is the emission intensity of the N, F-CDs@Cu^2+^ in the absence of S^2−^. The R^2^ coefficient of 0.99938, from 0 to 200 μM, was estimated by linear fitting, and the LOD was calculated to be 347 nM using the equation 3σ/k. Table 2 illustrates a comparison of the present study with previously reported systems, suggesting that the N, F-CDs@Cu^2+^ system has a wider linear range with a lower detection limit, implying the N, F-CDs@Cu^2+^ can detect S^2−^, specifically, with high accuracy.

### 2.5. Sensing Mechanism for the Detection of Cu^2+^ and S^2−^

The fluorescence sensing mechanisms are essential for the detection system to explore the interactions between ions and CDs. Hence, the chemical and optical properties were further investigated to investigate the sensing mechanism of the N, F-CDs. First, Figure 7a shows the relationship between the absorption spectra of Cu^2+^ (blue curve) and the excitation (black curve) and emission (red curve) of N, F-CDs, which were used to analyze IFE and FRET. The ultraviolet absorption band of the Cu^2+^ solution did not overlap with the excitation or emission spectrum of the N, F-CDs, thus excluding the possibility of the IFE or FRET being the fluorescence sensing mechanism of N, F-CDs. Subsequently, the UV-vis absorption spectrum was used to analyze the sensing mechanism. Static quenching forms stable complexes, resulting in peak shifts in the UV-vis absorption spectrum. Figure 7b illustrates a comparison of the absorption of the N, F-CDs and the N, F-CDs@Cu^2+^. The absorption peak at 230 nm did not shift with the addition of Cu^2+^, indicating that the static quenching was not the sensing mechanism. Additionally, the fluorescence lifetime decay spectra of the N, F-CDs (Figure 7c), under different conditions, were recorded to verify whether dynamic quenching is the sensing mechanism. All fluorescence decay spectra were fitted with a bi-exponential curve. The average fluorescence lifetime of the N, F-CDs was 5.05 ns, while the lifetime of N, F-CDs, in the presence of 200 μM Cu^2+^, was calculated to be 5.03 ns. Remarkably, the lifetimes of the N, F-CDs did not significantly change after the addition of Cu^2+^. This result implies dynamic quenching or PET is not the sensing mechanism, which causes a significant change in the lifetime of the N, F-CDs. The XPS spectrum was used to investigate the functional groups of N, F-CDs@Cu^2+^, and Figure 7d shows the high-resolution N1s when Cu^2+^ was added. Three peaks were observed at 399.1, 400.18, and 401.3 eV, which are assigned to pyridinic N, Cu–N, and graphitic N, respectively. Notably, the content of pyridinic nitrogen decreased while that of graphitic nitrogen increased, indicating the occurrence of a redox reaction. Simultaneously, Cu^+^ ions formed a new bond with nitrogen, implying that Cu^+^ entered the N, F-CD structure. A high-resolution XPS profile was used to further demonstrate the fluorescence sensing mechanism of the N, F-CDs in the presence of Cu^2+^. The high-resolution Cu1s spectrum (Figure 7e) exhibited three peaks at 932.62, 933.60, and 952.50 eV. The peaks at 932.62 and 933.60 eV are ascribed to Cu^2+^ and that at 952.50 eV is attributed to Cu^+^, indicating the partial reduction in Cu^2+^ to Cu^+^. Therefore, a redox reaction is suggested as the possible sensing mechanism that caused the fluorescence quench of the N, F-CDs. The high-resolution S1s spectrum of N, F-CDs@Cu^2+^ + S^2−^ was used to determine the sensing mechanism of S^2−^. As shown in Figure 7f, the two peaks at 161.68 and 162.88 eV correspond to CuS and Cu_2_S, respectively. In addition, the peak at 168.58 eV is attributed to NaS_2_O_3_, which is caused by the hydrolysis of S^2−^ ions. These results indicate that quenching is further caused, owing to the high affinity of Cu^+^ for S^2−^, after the S^2−^ addition to the N, F-CDs@Cu^2+^ sensing system.

### 2.6. Detection of Cu^2+^ and S^2−^ in Real Water Samples

Due to the development of the chemical industry and the imperfection of sewage treatment systems, copper and sulfide will also exist in the environment. For this reason, evaluating the accuracy and quantification of the fluorescence sensors in real water samples (tap water and local lake water) is very significant. The practicality of the proposed probes was verified by the spike–recovery method. Table 3; Table 4 show that, with different spiked concentrations of Cu^2+^ and S^2−^ in water samples, recovery rates in the tap and lake water samples ranged from 97.8 to 102.8% and 97.7 to 103.1%, respectively, indicating the complex conditions in real samples had no effect on the accuracy of the detection. These results show that the N, F-CDs can reliably and quantitatively detect Cu^2+^ and S^2−^ in real water samples, which demonstrates the great potential of the N, F-CDs for practical applications.

## 3. Materials and Methods

### 3.1. Reagents

2,4-difluorobenzoic acid, L-lysine, metal cation compound, and sodium salts (CuSO_4_, MgCl_2_·6H_2_O, CaCl_2_, ZnCl_2_, MnCl_2_, KI, (CH_3_COO)_2_Pb, (CH_3_COO)_2_Sr, (CH_3_COO)_2_Ba, LiCl, CdCl_2_, AgNO_3_, CsBr, NaCl, NaF, KHF_2_, NaBr, Na_2_SO_4_, Na_2_SO_3_, Na_2_CO_3_, NaHCO_3_, NaNO_3_, NaNO_2_, NaH_2_PO_4_, and NaH_2_PO_2_) were purchased from Macklin Biochemical Co., Ltd. (Shanghai, China) and of at least analytical grade. Deionized water was the solvent of all of the solutions, and all of the reagents were used directly without further treatment.

### 3.2. Apparatus

Transmission electron microscopy (TEM, FEI Tecnai G2 F20, FEI Company, College Station, TX, USA), X-ray photoelectron spectroscopy (XPS, Thermo Fisher ESCALAB 250Xi, Waltham, MA, USA), and Fourier-transform infrared spectroscopy (FT-IR, Nicolet 5700 spectrometer, Madison, WI, USA) were employed to reveal the morphology, the chemical composition, and the chemical structures of the N, F-CDs, respectively. Ultraviolet-visible spectroscopy (UV-vis, UV-2550 Shimadzu, Kyoto, Japan) was employed to determine the absorption of CDs and chemicals. Fluorescence spectroscopy, fluorescence lifetime decay spectra, and quantum yield (Fluorolo^@^-3 steady-state spectrofluorometer, HORIBA Scientific, Kyoto, Japan) were recorded to investigate the optical properties of the N, F-CDs.

### 3.3. Preparation of N, F-CDs

The N, F-CDs were prepared via a hydrothermal method, using 2,4-difluorobenzoic acid as the carbon and fluorine source and L-lysine as the nitrogen source. Here, 2 g 2,4-difluorobenzoic acid and 2 g L-lysine were dissolved in 20 mL deionized water, in a Teflon-lined autoclave (50 mL), and heated at 180 °C for 12 h. After cooling down to 25 °C, the supernatant of the reaction product was filtered using a 0.22-μm micropore film filter to remove small impurities and yield an N, F-CD solution. The N, F-CD solution was dialyzed for 24 h and then freeze-dried to obtain N, F-CDs powder for the subsequent weighing and characterization. The weight of N, F-CDs were determined to be 2.5 mg mL^−1^, and the aqueous N, F-CDs solution (0.25 mg mL^−1^) was used for subsequent experiments.

### 3.4. Detection of Cu^2+^ Ions

The quenching effect of Cu^2+^ ions on N, F-CDs fluorescence: N, F-CD solution (200 μL, 0.25 mg mL^−1^), deionized water (2175 μL), and the ions related to environment (125 μL, 10 mM) were mixed in a 4.5-mL quartz cuvette. The fluorescence spectra were recorded after 10 min of reaction to evaluate the selectivity of the fluorescent probes. Additionally, N, F-CD solutions (200 μL, 0.25 mg mL^−1^), deionized water (2200 μL), and the Cu^2+^ solution of different concentrations (100 μL) were mixed in a 4.5-mL quartz cuvette. The fluorescence spectra were recorded under 468 nm excitation, after 10 min of reaction, to plot a standard Cu^2+^-detection curve to evaluate the sensitivity of the fluorescent probes.

### 3.5. Detection of S^2−^ Ions

The N, F-CDs@Cu^2+^ sensing platform was formed by mixing 2150, 200, and 50 μL of deionized water, N, F-CD solutions (0.25 mg mL^−1^), and Cu^2+^ solutions (10 mM), respectively, in a quartz cuvette and incubating for 5 min. Subsequently, different concentrations of S^2−^ solutions (100 μL) were added to the N, F-CDs@Cu^2+^ sensing platform, and they recorded the fluorescence spectra, after 10 min of reaction, to evaluate the sensitivity of the N, F-CDs@Cu^2+^ sensing platform. Moreover, various ionic solutions were added to the detection system to investigate the selectivity of the N, F-CDs@Cu^2+^ sensing platform towards S^2−^ ions.

### 3.6. Detection of Cu^2+^ and S^2−^ in Real Water Samples

Tap water and lake water (Shenzhen, China), filtered through a 0.22-μm polyethersulfone water-phase membrane, were used as the real water samples. Different concentrations of solutions for detection were obtained by mixing the filtered water with Cu^2+^ and S^2−^ solutions. Subsequently, the N, F-CDs and N, F-CD@Cu^2+^ were added to these solutions and stood for 5 min, and then, the fluorescence emission intensity was recorded to quantify the contents of Cu^2+^ and S^2−^. Triplicate all experiments for reducing the experimental error.

## 4. Conclusions

In summary, environmentally friendly nitrogen and fluorine co-doped CDs with green emission were synthesized, using 2,4-difluorobenzoic acid and L-lysine as precursors, via a hydrothermal method. The obtained N, F-CDs exhibited a significant quantum yield and was covered with numerous surface functional groups that could be employed for Cu^2+^ detection. Based on the redox reactions, the N, F-CDs achieved quantitative Cu^2+^ detection, with a linear range of 0–200 μM and a detection limit of 215 nM. In addition, the formation of more stable nanocomplexes, due to the high affinity of Cu^2+^ for S^2−^, further quenched the fluorescence of N, F-CDs@Cu^2+^, which could be used to detect S^2−^, achieving a linear range of 0–200 μM with a detection limit of 347 nM. The N, F-CDs were used to measure the contents of Cu^2+^ and S^2−^ in real water samples and demonstrated satisfactory recoveries of 97.8~102.8% and 97.7~103.1%, respectively. These results suggest that the proposed fluorescence probes have high prospects for practical applications in the detection of Cu^2+^ and S^2−^.

## Figures and Tables

**Figure 1 molecules-27-05149-f001:**
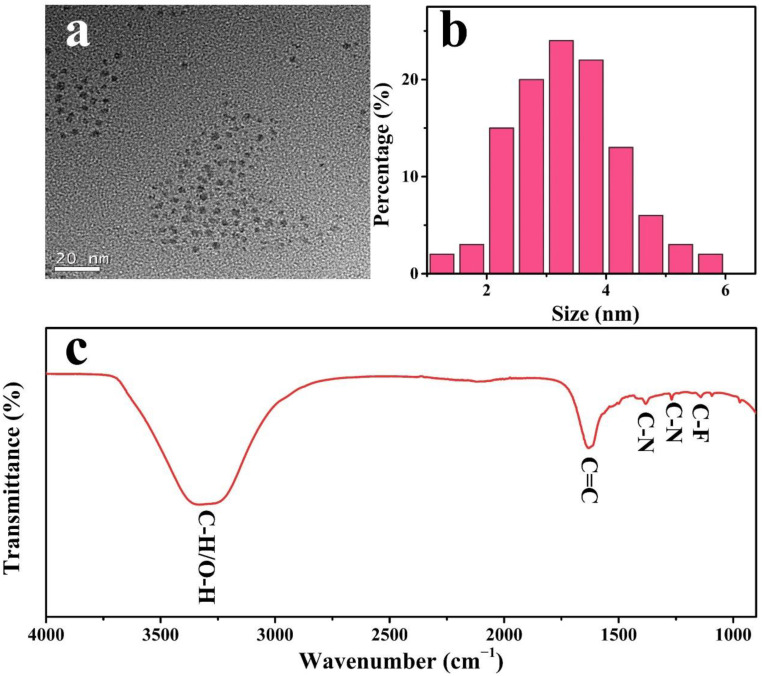
(**a**) TEM image of the N, F-CDs. (**b**) Particle size distribution of the N, F-CDs. (**c**) FT-IR spectrum of the N, F-CDs.

**Figure 2 molecules-27-05149-f002:**
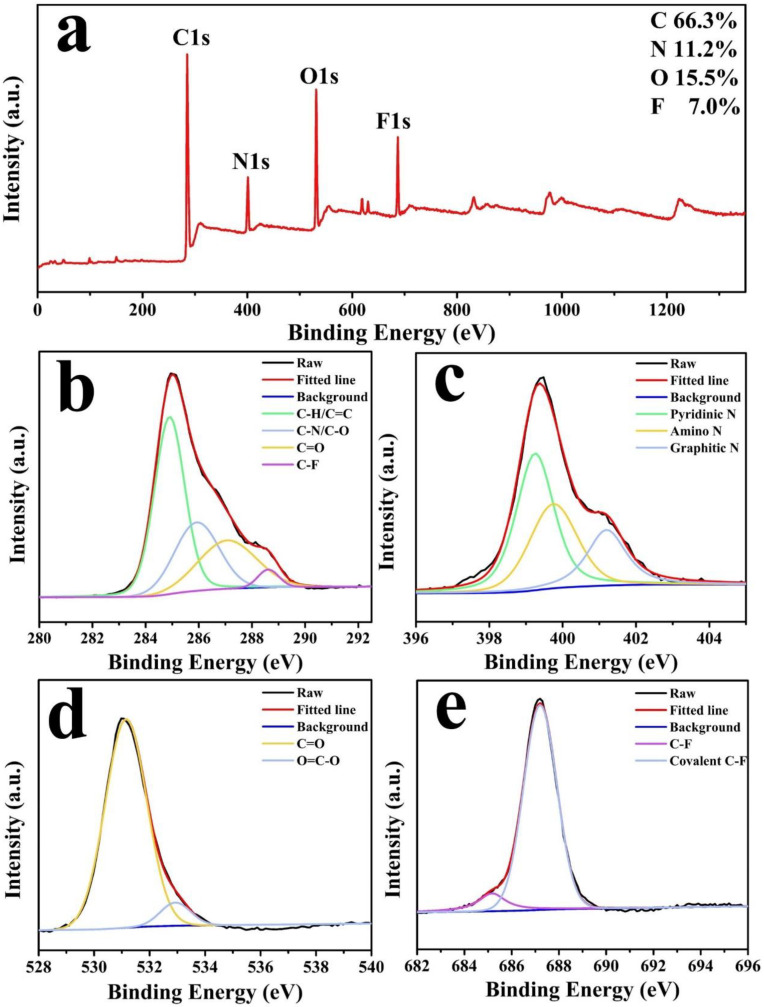
(**a**) XPS full profiles of N, F-CDs. High-resolution XPS spectrum of: (**b**) C1s. (**c**) N1s. (**d**) O1s. (**e**) F1s.

**Figure 3 molecules-27-05149-f003:**
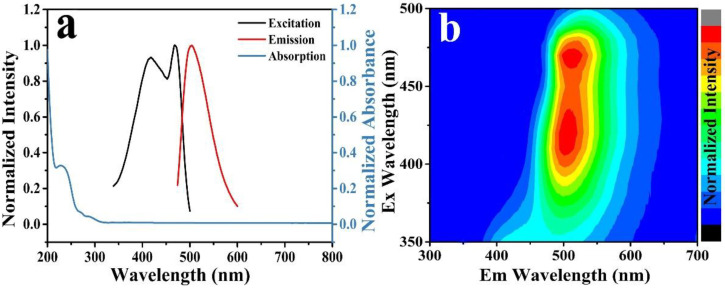
(**a**) UV-vis absorption (blue line), fluorescence excitation (black line), and emission spectra (red line) of the N, F-CDs in an aqueous solution. (**b**) The excitation-emission map of the N, F-CDs aqueous solution in the excitation range of 300–600 nm.

**Figure 4 molecules-27-05149-f004:**
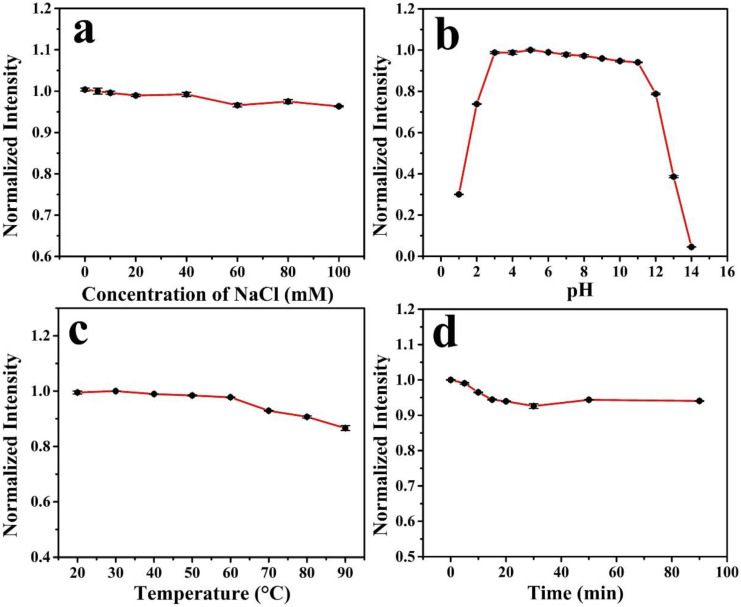
(**a**) The effect of various NaCl concentrations on the N, F-CDs fluorescence intensity. (**b**) The effect of pH on the fluorescence intensity of the N, F-CDs. (**c**) Normalized fluorescence intensity at different temperature. (**d**) Photostability of the N, F-CDs.

**Figure 5 molecules-27-05149-f005:**
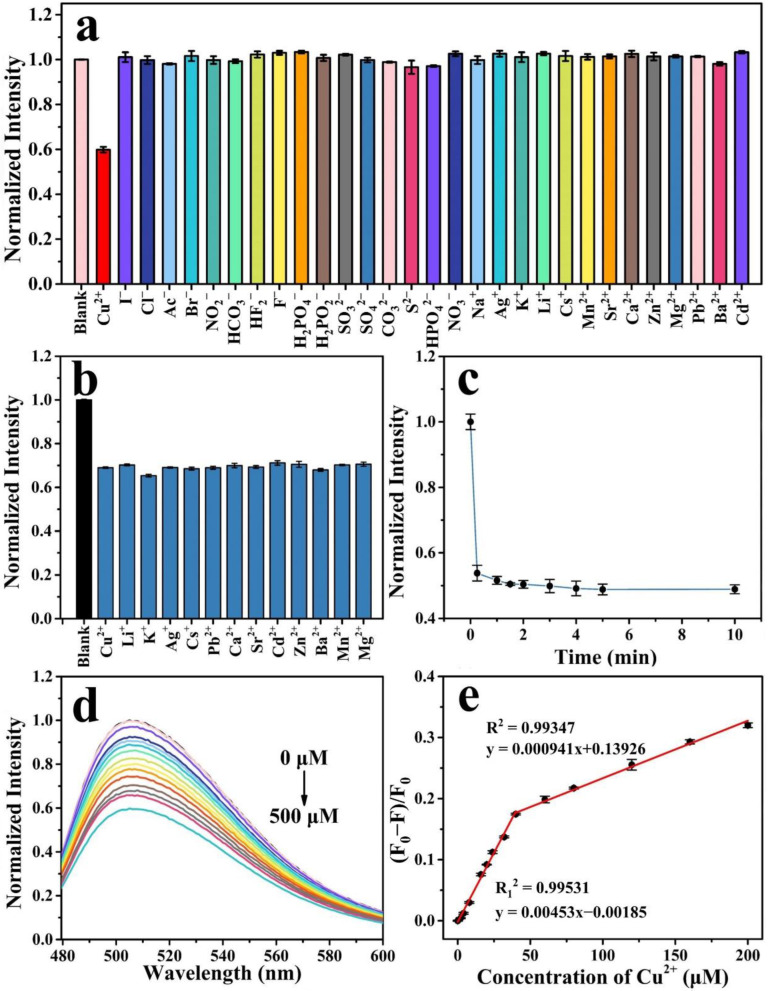
(**a**) Fluorescence response of the N, F-CDs to the various metal ions and anions at a concentration of 500 μM. (**b**) Anti-interference ability of the N, F-CDs in Cu^2+^ detection. (**c**) Reaction time after the addition of Cu^2+^. (**d**) Fluorescence emission spectra of the N, F-CDs at different Cu^2+^ concentrations (from top to bottom: 0–500 μM). (**e**) The linear relationship between the ratio of fluorescence intensity ((F_0_ − F)/F_0_) changes and Cu^2+^ concentrations (0–200 μM).

**Figure 6 molecules-27-05149-f006:**
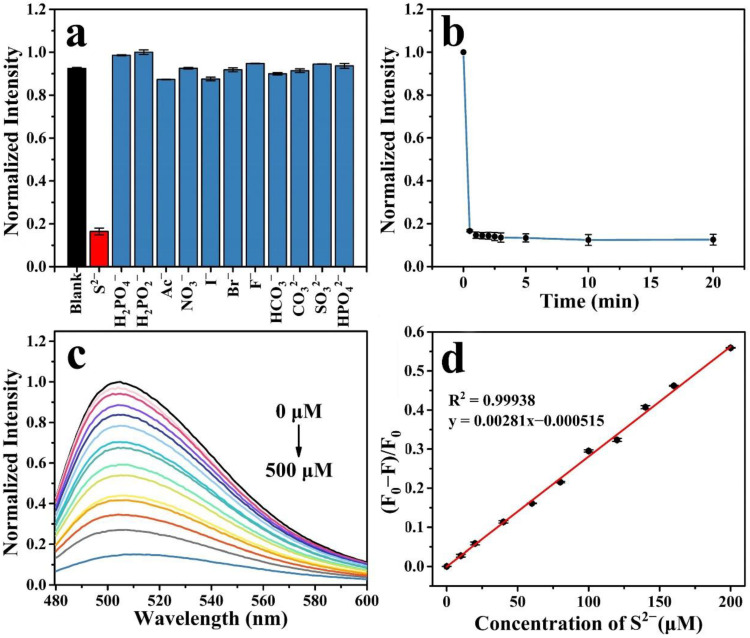
(**a**) Fluorescence response of N, F-CDs@Cu^2+^ to the various anions at a concentration of 500 μM. (**b**) Reaction time after the addition of S^2−^. (**c**) Fluorescence emission spectra of N, F-CDs@Cu^2+^ at different S^2−^ concentrations (from top to bottom: 0–500 μM). (**d**) Linear relationship between the ratio of fluorescence intensity ((F_0_ − F)/F_0_) changes with S^2−^ concentrations (0–200 μM).

**Figure 7 molecules-27-05149-f007:**
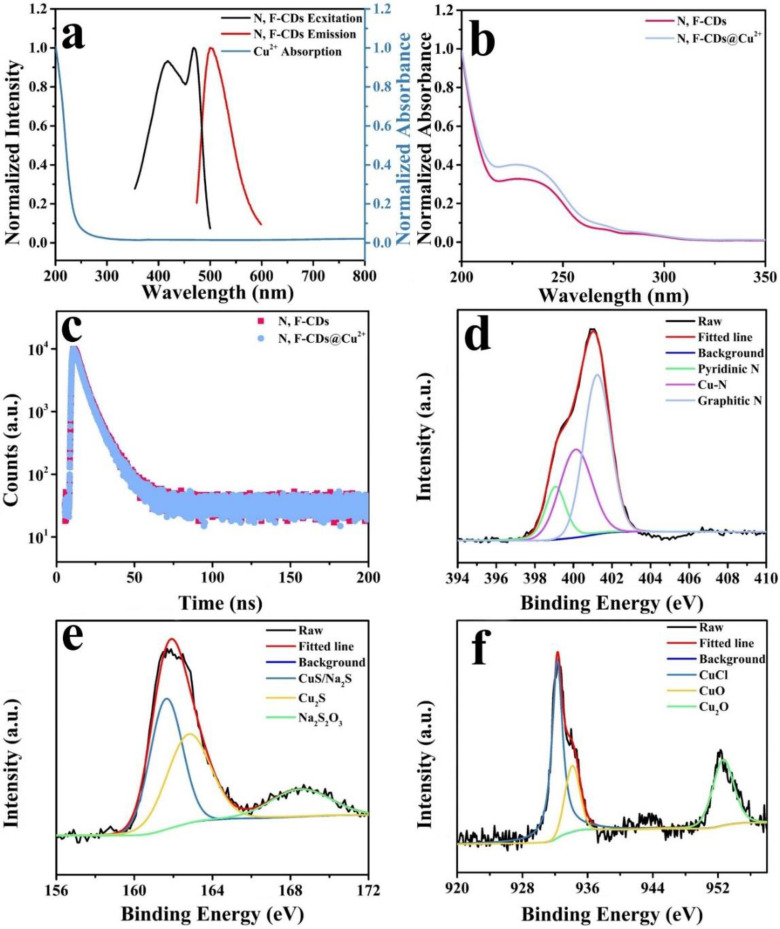
(**a**) UV absorption of Cu^2+^ (blue line), excitation (black line), and emission (red line) spectra of the N, F-CDs. (**b**) UV absorption of the N, F-CDs and N, F-CDs@Cu^2+^. (**c**) Spectra of the fluorescence lifetime decay of the N, F-CDs and N, F-CDs@Cu^2+^. (**d**) High-resolution N1s XPS profiles of the N, F-CDs@Cu^2+^. (**e**) High-resolution Cu1s XPS profiles of the N, F-CDs@Cu^2+^. (**f**) High-resolution S1s XPS profiles of the N, F-CDs@Cu^2+^ + S^2−^.

**Table 1 molecules-27-05149-t001:** Comparison of different optical methods applied for the determination of Cu^2+^.

Detection Condition	Linear Range (μM)	Detection Limit (μM)	Application	Reference
Carbon nanosheets	20–40	4.9	Lake water	[35]
d-CDs	0.8–55	0.043	Tea, Raisin, Kiwifruit	[36]
PEI-DA	0–30	0.193	Cell monitor	[37]
Fe_3_O_4_@AP-B(OH)_2_ nanocomposite	1–30	0.3	-	[38]
Gold nanoparticles	0–108	5.8	Lake water	[39]
N, F-CDs	0–200	0.215	Lake water, Tap water	This work

**Table 2 molecules-27-05149-t002:** Comparison of different optical methods applied for the determination of S^2−^.

Detection Condition	Linear Range (μM)	Detection Limit (μM)	Application	Reference
F-SiNPs	0–100	0.1	Cell imaging	[40]
TSOC-Cu^2+^	0–60	0.362	Cell imaging	[41]
CQDs-O-NBD	0–10	0.18	Cell imaging	[42]
N-doped CDs	0.05–10	0.032	Lake water	[43]
FCD-Cu^2+^	0–10	0.089	Cell imaging	[44]
N, F-CDs@Cu^2+^	0–200	0.347	Lake water, Tap water	This work

**Table 3 molecules-27-05149-t003:** Determination of Cu^2+^ in lake water and tap water. (*n* is the number of repeated experiments, 95% confidence level, RSD = recovery standard deviation).

Sample	Spiked Cu^2+^ (μM)	Detected (μM)	Recovery (%)	RSD (*n* = 3) (%)
Lake water	1	0.99	99.3	1.44
	2	1.96	97.8	0.85
	4	4.11	102.8	1.30
Tap water	1	1.02	102.1	0.33
	2	2.06	103	0.95
	4	3.99	99.8	1.58

**Table 4 molecules-27-05149-t004:** Determination of S^2−^ in lake water and tap water. (*n* is the number of repeated experiments, 95% confidence level, RSD = recovery standard deviation).

Sample	Spiked S^2−^ (μM)	Detected (μM)	Recovery (%)	RSD (*n* = 3) (%)
Lake water	1	1.02	102.0	1.60
	2	2.01	100.4	1.21
	4	3.94	98.6	1.46
Tap water	1	0.98	97.7	1.88
	2	2.06	103.1	3.89
	4	4.03	100.8	2.29

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
