# Peer review of "Bifunctional Nitrogen and Fluorine Co-Doped Carbon Dots for Selective Detection of Copper and Sulfide Ions in Real Water Samples"

_molecules, 2022, doi:10.3390/molecules27165149_

Round 1

Reviewer 1 Report

Although the manuscript “Bifunctional nitrogen and fluorine co-doped carbon dots for selective detection of copper and sulfide ions in real water samples” could be an interesting paper I think that it is not suitable to be published in Molecules in the present form. The materials applied for the proposed method is not an original matter, however the methodology could be significant for the detection of copper and sulfur ion in the environmental samples.

Some specific comments:

Line 16 to 18: these two sentence should be rechecked to be correct and comprehensible.

Line 40-42: the information of the methods used for the determination of copper and sulfur should be rechecked to be more precise.

Line 212: Figure 5.(e) R squared instead of R2 squared

Line 283: actual water samples or real water samples, please clarify and present it consistently in the text.

Line 295: Table 3, please show the concentrations of copper ion in lake water and tape water without the spike of copper ion

Line 297: Table 4, please show the concentrations of sulfur ion in lake water and tape water without the spike of sulfur ion

The precision of the proposed methods should be indicated with detail.

Author Response

Response to the Referees’ Comments

We thank the referees’ positive and valuable comments. The following are point-by-point to the referees’ comments/questions.

Reviewer #1:

Although the manuscript “Bifunctional nitrogen and fluorine co-doped carbon dots for selective detection of copper and sulfide ions in real water samples” could be an interesting paper I think that it is not suitable to be published in Molecules in the present form. The materials applied for the proposed method is not an original matter, however the methodology could be significant for the detection of copper and sulfur ion in the environmental samples.

Some specific comments:Line 16 to 18: these two sentence should be rechecked to be correct and comprehensible.Response 1:

We do appreciate your positive and valuable comments. We have modified the text and the revised part is shown as follows:

The N, F-CDs based on the redox reactions for selective and quantitative detection of Cu2+ showed a wide linear range (0–200 μM) with a detection limit (215 nM). By forming the N, F-CDs@Cu2+ sensing platform and based on the high affinity of S2- to Cu2+, the N, F-CDs@Cu2+ can specifically detect S2- over a linear range of 0–200 μM and a detection limit of 347 nM.

Line 40-42: the information of the methods used for the determination of copper and sulfur should be rechecked to be more precise.

Response 2:

We do appreciate your positive and valuable comments. We have modified the text and the revised part is shown as follows:

Current, studies have applied techniques, such as ion chromatography [17], and spectrophotometric titration [18] for Cu2+ detection; capillary electrophoresis [19-22] and chemodosimeter [23] for S2- detection. However, these methods require multi-step preprocessing, and have a high cost and low selectivity.

  1. Ferreira, N.L.; de Cordova, L.M.; Schramm, A.D.S.; Nicoleti, C.R.; Machado, V.G. Chromogenic and fluorogenic chemodosimeter derived from Meldrum's acid detects cyanide and sulfide in aqueous medium. J Mol Liq 2019, 282, 142-153, doi:10.1016/j.molliq.2019.02.129.

Line 212: Figure 5. (e) R squared instead of R2 squared.

Response 3:

Thanks for your careful review. We have modified the Fig 3a and 3b, and the revised figure is shown as follows:

Line 283: actual water samples or real water samples, please clarify and present it consistently in the text.

Response 4:

Thanks for your careful review. We have modified the text and the revised part is shown as follows:

3.6. Detection of Cu2+ and S2- in real water samples.

Line 295: Table 3, please show the concentrations of copper ion in lake water and tape water without the spike of copper ion

Line 297: Table 4, please show the concentrations of sulfur ion in lake water and tape water without the spike of sulfur ion

Response 5:

We do appreciate your positive and valuable comments.

Before the spike recovery method, we conducted a comparison experiment between blank samples and real water samples. The blank sample consisted of N, F-CDs + deionized water, while the real sample was N, F-CDs + lake/tap water. Comparing their fluorescence intensities, it is found that the fluorescence intensity does not change in real water samples. Based on these experiments, the real water samples will not affect the fluorescence intensity of the N, F-CDs. In other words, copper and sulfide ion could not be found in real water samples.

The precision of the proposed methods should be indicated with detail.

Response 6:

We do appreciate your positive and valuable comments. We have provided more details about the proposed methods and modified the revised part is shown as follows:

Table 1. Comparison of different optical methods applied for the determination of for Cu2+.

Detection condition

Linear range

(μM)

Detection limit

(μM)

Application

Reference

Carbon nanosheets

20 - 40

4.9

Lake water

[35]

d-CDs

0.8 - 55

0.043

Tea, Raisin, Kiwifruit

[36]

PEI-DA

0 - 30

0.193

Cell monitor

[37]

Fe3O4@AP-B(OH)2 nanocomposite

1 - 30

0.3

-

[38]

Gold nanoparticles

0 - 108

5.8

Lake water

[39]

N, F-CDs

0 - 200

0.215

Lake water, Tap water

This work

Table 2. Comparison of different optical methods applied for the determination of S2-.

Detection condition

Linear range

(μM)

Detection limit

(μM)

Application

Reference

F-SiNPs

0 - 100

0.1

Cell imaging

[40]

CDs-Ag NCs

0.5 - 100

0.35

Mineral water, Tap water

[41]

CQDs-O-NBD

0 - 10

0.18

Cell imaging

[42]

N-doped CDs

0.05 - 10

0.032

Lake water

[43]

FCD-Cu2+

0 - 10

0.089

Cell imaging

[44]

N, F-CDs@Cu2+

0 - 200

0.347

Lake water, Tap water

This work

Reviewer 2 Report

Please find the attachment

Author Response

Response to the Referees’ Comments

We thank the referees’ positive and valuable comments. The following are point-by-point to the referees’ comments/questions.

Reviewer #2:

Review of the manuscript “Bifunctional nitrogen and fluorine co-doped carbon dots …” by Yutian Zeng e.a.

Nitrogen- and fluorine-containing carbon dots were synthesized by a simple protocol and applied for the determination of copper (2+) and (indirectly) sulfide ion by a very simple procedure. The selectivity is high, and the detection limits are comparable to those obtained with the other fluorescent probes. Overall, the manuscript leaves a good impression.

Major concerns and questions:

  1. It is important to incorporate F atoms into the carbon dots and why? Please explain.

Response 1:

We truly thanks for your valuable comments.

As an electron withdrawing group, F acts as a ligand on the surface of N, F-CDs, and because of its strong electronegativity and its great characteristic that can form hydrogen bonds easily, it can optimize the optical properties of N, F-CDs,improve the fluorescence stability and biocompatibility. After we refer to the relative report (Molecules 2022, 27(14), 4620), fluorine might alter the energy gaps between the highest occupied and lowest unoccupied molecular orbitals of the N, F-CDs and lead to a higher quantum yield.

  1. Fig. 3: the excitation maximum is at 400–450 nm, while there is no absorption in this region – how can it happen? The same for Fig. 7a.

Response 2:

We do appreciate your positive and valuable comments.

The structure of carbon dots cannot be precisely determined, which is different from the organic molecules. We refer to the relative works, e.g., Spectrochimica Acta Part A: Molecular and Biomolecular Spectroscopy 2022, 281, 121597; Applied Surface Science 2022, 599, 153705. The excitation and absorption spectrum in these works are shown below. The excitation and absorption peaks do not overlap significantly. The luminescence of carbon dots might be determined by the combination of carbon nucleus and surface functional groups, so their absorption peak is no in region of their excitation peak.

Minor remarks:

Line 42: “However, these methods require multi-step pre-pressing, and have a high cost and low selectivity” – does it mean “preprocessing”?

Response 3:

Thanks for your careful review. We have modified the text and revised part is shown as follows:

  However, these methods require multi-step preprocessing, and have a high cost and low selectivity.

C–H stretching band is not likely to be at 3300 cm–1 , is it supposed to be at 3000–3100 cm–1.

Response 4:

We do appreciate your positive and valuable comments.

In the FT-IR spectrum, 3328 cm–1 is just the maximum of this broad peak. The broad peak range from 3725 to 2765 cm–1, which includes the C-H stretching vibration. On the other hand, 2,4-difluorobenzoic acid was used as a precursor for synthesis of N, F-CDs. However, due to the existence of the benzene ring, the C–H stretching vibration absorption peak would appear at higher wavenumbers. So, the C–H stretching band appears at 3328 cm–1 is reasonable.

Tables 1 and 2 should be named “…optical methods…” (or fluorescence-based).

Everywhere: absorptions → absorption.

Line 283: 3.6. “Detection of Cu2+ and S2- in actual water samples” = “…in real water samples”.

In ref. 40 the year and volume are missing

Response 5:

Thanks for your careful review and good suggestions. We have modified these presentations according to your suggestions and the ref. 40 have modified as follows:

  1. Yang, J.; Huang, Y.; Cui, H.Y.; Li, L.; Ding, Y.P. A FRET Fluorescent Sensor for Ratiometric and Visual Detection of Sulfide Based on Carbon Dots and Silver Nanoclusters. JOURNAL OF FLUORESCENCE 2022, 32, doi:10.1007/s10895-022-02981-8.
